# Uses of *Polypodium leucotomos* Extract in Oncodermatology

**DOI:** 10.3390/jcm12020673

**Published:** 2023-01-14

**Authors:** Paolo Calzari, Silvia Vaienti, Gianluca Nazzaro

**Affiliations:** 1Department of Pathophysiology and Transplantation, University of Milan, 20122 Milan, Italy; 2Section of Dermatology and Venereology, Department of Medicine, University of Verona, 30127 Verona, Italy; 3Dermatology Unit, Foundation IRCCS, Ca’ Granda Ospedale Maggiore Policlinico, Via Pace 9, 20122 Milan, Italy

**Keywords:** chemoprevention, cancer, cell death, metabolism, cell invasion, metastasis

## Abstract

The effects of UV radiation on the skin and its damage mechanisms are well known. New modalities of exogenous photoprotection have been studied. It was demonstrated that *Polypodium leucotomos* extract acts as an antioxidant, photoprotectant, antimutagenic, anti-inflammatory, and immunoregulator. It is effective when taken orally and/or applied topically to support the prevention of skin cancers. It also has an important role in preventing photoaging. This review aims to report the mechanisms through which *Polypodium leucotomos* acts and to analyze its uses in oncodermatology with references to in vitro and in vivo studies. Additionally, alternative uses in non-neoplastic diseases, such as pigmentary disorders, photosensitivity, and atopic dermatitis, have been considered.

## 1. Introduction

The sun emits a wide spectrum of electromagnetical waves, of which ultraviolet light (UV) is the most aggressive against cellular compounds. UV radiation (UVR) is both a mutagen and a non-specific damaging agent. Depending on the wavelength, UV rays are divided into UVA (320–400 nm), UVB (280–320 nm), and UVC (200–280 nm) [1].

Ozone screens out significant quantities of UVB and UVC. Therefore, solar UV radiation reaching the skin consists of 5–10% highly energetic UVB (290–315 nm) and 90–95% UVA (315–400 nm). UVA radiation is less energetic, but because of its longer wavelength, it penetrates deeper into the cutaneous layers [2].

### UV Damage Occurs *via* Different Mechanisms

UVA radiation produces reactive oxygen species (ROS) by interacting with endogenous photosensitizers. ROS cause indirect damage to DNA, proteins, membranes, dermal connective tissue, and proteins. Furthermore, UVA radiation induces the formation of 8-deoxyguanosine (8-OH-dG) [3], a product of DNA oxidation, and deletions in mitochondrial DNA [4].

On the other hand, the absorption of UVB photons leads to the disruption of DNA (“UV signatures” [5]), resulting in cyclobutane pyrimidine dimers (CPD) and pyrimidine (6-4) pyrimidone photoproducts (6-4PP). Indeed, due to its aromatic heterocyclic bases, DNA is a strong UVB chromophore (absorption maximum at 260–265 nm). The cellular production of CPDs continues four hours after the end of UV exposure (dark-CPDs). Mutations in the genes that play a central role in tissue homeostasis and genome integrity can occur as a result of DNA damage accumulation [6].

Other negative effects of UVR include inflammation, immunosuppression, and the remodeling of the extracellular matrix (ECM).

Inflammation is due to vasodilatation mediated by nitric oxide and prostaglandins, and it leads to the recruitment of macrophages and neutrophils. Additionally, UVR is able to act on Langerhans cells and mastocytes [7,8].

Immunosuppression is mediated by cis-urocanic acid (c-UCA) [4]. C-UCA derives from the isomerization, inducted by photon absorption, of trans-urocanic acid (t-UCA), the main product from histidine metabolism with photoprotective and ROS scavenging properties [9,10,11]. Large quantities of c-UCA act on epidermal Langerhans cells and induce the abnormal degranulation of mastocytes [12,13]. Thus, the isomerization of t-UCA to c-UCA converts UV radiation into an immunosuppressive signal [14]. In addition, the immunosuppressive effects of UVR are mediated by immunomodulatory molecules, including PGE2, TNF-α and IL-10 [15].

The remodeling of the ECM is due to the upregulation of matrix metalloproteinases (MMPs) and decreased collagen synthesis (in particular, type I [16]), inhibitors of matrix metalloproteinases (TIMPs), and transforming grow factor-β (TGF-β) [17,18,19].

Hence, both UVA and UVB play a crucial role in photodamage (photocarcinogenesis and photoaging) [18,20], photosensitivity [21], and disorders of pigmentation [22,23]. Emerging evidence has shown a possible role of infrared (IR) [24] and blue light [25] in photoaging.

Photoprotection reduces the aforementioned damaging effects of UVR. It is classically categorized into three modalities, which are not mutually exclusive:

(1) Physical protections (hats, sunglasses and clothes);

(2) Topical sunscreens (physical barriers that reflect and scatter light, and chemical barriers, that absorb light);

(3) Systemic photoprotectors (antioxidants, osmolytes, and DNA repair enzymes) [26,27].

Systemic photoprotective agents appear to act through their anti-inflammatory and antioxidant capacities. They represent an emerging and most-studied field in dermatology. They provide not only an additional photoprotection at the time of the exposition but also likely revert the damage of previous UV exposure. Thus, they can be suggested as an additional method for UV radiation protection [26].

Most systemic photoprotectors and some chemical filters are phytochemicals. Some of these can be obtained naturally from native sources. Among the latter, *Polypodium leucotomos* extract (PLE) has been studied and introduced on the market, alone or in association, both in oral supplements and sunscreens [28].

This review aims to report the mechanisms through which *Polypodium leucotomos* (PL) acts and analyses its uses in oncodermatology. Lastly, alternative uses in non-neoplastic diseases have been considered.

## 2. Origin and Chemical Composition of *Polypodium leucotomos* Extract

PL is a fern native to Central and South America, where it has a historical role in traditional medicine [29]. PL belongs to the Polypodiaceae family, genus *Phlebodium.* The concentrated hydrophilic extract of the leaves of *Polypodium leucotomos,* endowed with photoprotective properties, is marketed with the commercial name Fernblock [30].

PLE is made up of phenolic compounds (benzoates and cinnamates) and biological acid molecules (quinic, shikimic, glucuronic, and malic acids). Phenolic compounds are plant metabolites found in fruits, vegetables, coffee beans, and tea leaves [30]. Among acid molecules, the most represented are 4-hydroxycinnamic acid (p-coumaric), 3-methoxy-4-hydroxycinnamic acid (ferulic), 3-dihidroxycinnamic acid (caffeic), 3-methoxy-4-hydroxybenzoic acid (vanillic), and 3-caffeoliquinic acid (chlorogenic) [4,31].

PLE provides both topical and systemic protection due to its chemical composition [4].

## 3. Pharmacodynamic

### 3.1. Direct Antioxidant Activity

The direct antioxidant activity is attributable to the polyphenolic compounds.

The antioxidant capacity of the single PLE component increases with concentration. In vitro, caffeic, chlorogenic, and ferulic acids were demonstrated to be the most potent inhibitors of oxidation, both for antioxidant function and the IC50 (the concentration at which oxidation is inhibited at 50%) value [30].

In particular, PLE inhibits membrane damage and lipid peroxidation in fibroblasts and keratinocytes [32].

Additionally, PLE acts as a direct ROS scavenger against superoxide anion, hydroxyl, oxygen, and oxygen peroxide [33,34,35]. It also acts by inhibiting glutathione oxidation [35].

This antioxidant activity of PLE was demonstrated by the reduced levels of 8-OH-dG, a marker of cellular oxidative damage [36].

Finally, an in vitro study demonstrated that the best antioxidant and photoprotective activity is obtained with an aqueous PLE extracted at basic pH [26].

### 3.2. Activity on DNA Damage and on Photocarcinogenesis

If administered orally, PL inhibits UV-mediated DNA damage and mutagenesis, reducing the accumulation of CPD [37]. This may be due to both the specific function of the DNA repair systems and the activity of oxidative damage [38].

Furthermore, PL induces the activation of the oncosuppressor p53 and prevents epidermal cell proliferation, decreasing the level of cyclin D1, proliferating cell nuclear antigen (PCNA) and Ki67 [34,37]. Nuclear factor kappa beta (NF-kB) and activator protein-1 (AP-1) are inhibited by PL, too [39].

Finally, PL decreases UV-induced mitochondrial DNA damage [40] and inhibits neoangiogenesis [38].

As a result, the cutaneous immune competence and tumor surveillance are restored, reducing the carcinogenic potential in photodamaged skin.

### 3.3. Immunomodulatory and Antinflammatory Activity

PLE inhibits the photoisomerization of trans-urocanic acid (t-UCA). Additionally, it was demonstrated that it inhibits its photodecomposition in the presence of oxidizing reagents, such as H2O2 and titanium dioxide (TiO2).

PLE inhibits pro-inflammatory cytokines, such as TNF-α and IL-12, inducible nitric oxide synthase (iNOS), mastocytes, neutrophils, and macrophages. Furthermore, it inhibits the apoptosis of dendritic cells and Langerhans cells [39,41,42,43].

COX-2 is responsible for prostaglandin synthesis. It has an important role in inflammation, but it is also involved in carcinogenesis. PLE decreases UV-induced COX-2 expression [36,41,42,43].

Moreover, ferulic and caffeic acid reduce UVB-induced erythema; this justifies their inclusion in several topical formulations [44].

### 3.4. Activity on Extracellular Remodeling

PLE can block UVR-induced apoptosis in fibroblasts and keratinocytes and re-establish their proliferation. Additionally, PL inhibits the disorganization of the actin cytoskeleton, cell–cell adhesive contacts, and cell matrix [45].

Furthermore, PLE acts on molecular targets, as well. It inhibits MMP-1,2,3 and 9. It stimulates TIMPs, TGF-β, elastin, and fibrillin. Finally, it stimulates the deposition of types I and V collagen in UV-irradiated fibroblasts and types I, III, and V collagen in non-irradiated fibroblasts [46,47].

## 4. Pharmacokinetics

When orally administered, PLE is absorbed rapidly and efficiently (70–100%). It is overall metabolized in 24 h in the liver. In particular, coumaric, phenolic, and vanillic acids are metabolized by monooxygenase enzymes dependent on cytochrome P450 (CYP450), suffering partial conjugation to glucuronic sulphate and glucuronic acid. No binding to cellular proteins is observed [30,48,49,50].

## 5. Formulation, Dosage and Safety

At the moment, PLE is available as a topical formulation (gel, cream, spray, and compact makeup powder) and as a systemic agent (oral capsules).

In Europe, it has been available as a supplement since 2001, and in the US since 2006. In the current formulation, it is marketed as a dietary supplement aimed at “protecting against sun-related effects and aging”, with a recommendation of one capsule of 240 mg in the morning. When extensive sun exposure is anticipated, the dosage of 240 mg is recommended one hour before the exposure and after 2–3 h. It has been available as a topical product in Europe since 2000 [51].

Finally, PLE has a good safety profile without side effects [52,53]. It has been shown to be safe in children [54], and no data are available concerning pregnancy [52].

## 6. Clinical Use in Photodamage Prevention

Skin photodamage is the long-term consequence of the cumulative effect of UVR exposure. It includes photocarcinogenesis and photoaging [55].

### 6.1. Photocarcinogenesis

Skin cancer is primarily caused by UVA radiation exposure. PLE demonstrated photoprotective and anti-inflammatory properties in several in vitro and in vivo studies, making it a potentially useful tool to prevent skin photocarcinogenesis [28,38,56,57].

#### 6.1.1. Oral PLE Supplementation

IN 2004, Middelkamp-Hup [58] studied the effects of oral supplementation with PLE in 10 healthy volunteers undergoing Psoralen UVA (PUVA) therapy.

All of the patients were exposed to PUVA (oral 8-MOP at 0.6 mg/kg; UVA source 320-400 nm) without receiving oral PLE. The back skin was exposed four times to the same photo test at 1 h, 1 h and 30 min, 2 h, and 2 h and 30 min to ensure optimal phototoxicity. The phototoxic reaction was evaluated 48 h after UVA exposure. Bioptical samples were taken from four patients after 48 h and from three patients after 72 h.

Later, the same PUVA treatment was repeated, and all of the patients received a single dose of oral PLE (7.5 mg/kg body weight) the evening before the procedure. Phototoxicity evaluations and biopsies were repeated. The authors demonstrated that PLE significantly decreases PUVA phototoxicity and hyperpigmentation. The histological examination of the sample collected without PLE administration displayed alterations in maturation, microvesiculation and the vacuolization of keratinocytes. Moreover, Langerhans cells increased in size and showed a loss of dendritic morphology. Contrarywise, in people treated with PLE, there was a lower grade of erythema and edema at all exposure time points (*p* < 0.005). Histologically, the number of sunburnt cells/mm epidermis was significantly lower (*p* < 0.05), as well as the depletion of Langerhans cells/mm2 epidermis (*p* < 0.01). The latter were also morphologically normal. Finally, there was a statistically significant reduction in tryptase-positive mast cells in the papillary dermis (*p* < 0.05) and of microvessels (*p* < 0.01), demonstrating how PLE leads to decreased PUVA vasodilatation.

In 2007, JanczykIn et al. [39] performed an in vitro study on human keratinocytes. An extract of hydrophilic PLE was used to pre-treat these cells. Then, the latter were irradiated with UVA and UVB rays (UVA = 6 J/cm^2^ + UVB = 0.5 J/cm^2^; and UVA = 11 J/cm^2^ + UVB = 1 J/cm^2^). The Authors found that PLE inhibits key factors implicated in UV-mediated skin carcinogenesis, such as the increase of TNF-α, the induction of NOS, the transcription of NFkβ and AP-1.

Zattra et al. [36] demonstrated that PLE reduces UV-induced COX-2 levels, promoting the expression of p53 in hairless mice. The researcher divided the mice into two groups. The first one was fed with PLE 300 mg/kg for 10 days; the second one did not receive PLE. Then, the mice were irradiated with UVB with a wavelength equal to 285 ± 5 nm in a single dose of 25 mJ/cm^2^. The authors assessed the increased p53 expression and the reduced cyclobutane pyrimidine photoproducts and COX-2 expression. After 24 h from the intake of PLE, the first group showed 79% fewer cyclobutane pyrimidine photoproducts positive cells (*p* < 0.03). Additionally, it showed reduced COX-2 levels (*p* < 0.05). The reduction in UV-induced mutations was also confirmed 2 weeks after the UV exposure.

Therefore, PLE reduces COX-2 levels in mice, promoting p53 expression and decreasing prostaglandin-mediated inflammation linked to photoaging. All of these effects have yet to be demonstrated in humans [59].

Another study was performed by Kholi et al. [37]. Twenty-two patients with Fitzpatrick skin types I–III were recruited. The subjects were irradiated on day 1 with visible light, ultraviolet A1 (UVA1), and ultraviolet B (UVB) using, respectively, 150 W EKE lamp, Hamamatsu LightingCure UV Spot Light 200, 200 W 240–400 nm, and a 308 nm excimer laser. Two skin biopsies for each participant were conducted 24 h after irradiation (day 2). One was conducted on unirradiated skin (control biopsy), and the other on an irradiated site in which was reached MED (minimal erythema dose). On day 3, the patients were treated with 240 mg of PLE 2 h and 1 h before irradiation. On day 4, the physician assessment and skin biopsies from the MED site were repeated. PLE significantly reduced clinical UVB alterations in 17 patients and induced histological changes in all patients. In the single patients, erythema intensity post-PLE was reduced by 8% (*p* < 0.05), as well the expression of several histological biomarkers associated with UVB damage (sunburn cells, COX2, Cyclin D1, Ki67, and proliferating cell nuclear antigen).

Finally, Portillo and his group in 2021 [60] conducted an in vitro study to demonstrate the efficacy of PLE against UV-induced effects on the skin. The authors used murine melanocytes (B16-F10) and exposed them to UVA radiation in order to produce dark-CPDs. Three hours after the irradiation, a significant amount of dark-CPD was observed. The PLE pretreatment inhibited this phenomenon. Finally, PLE also reduced the production of all reactive species, such as nitric oxide, superoxide, and peroxynitrite.

#### 6.1.2. Topical PLE Administration

Using a reconstructed human epidermis (Episkin^®^, SkinEthic Laboratories, Nice France), Torricelli et al. [61] conducted an in vitro study to evaluate the effects of topical PLE on UVB-induced cell damage. The authors performed a histological evaluation, a quantitative assessment of apoptotic sunburn cells, a measure of the thickness of the stratum corneum and an immunohistochemical evaluation. The topical application of 2 mg/cm^2^ of PL immediately before UVB exposure (300 mJ/cm^2^) reduced sunburn cells (80%), ki-67 (48%) positive cells, and cyclopyrimidine dimers (CPD). Moreover, an increase in p53 (80%) and p21 (84%) positive cells was observed, confirming the photoprotective effect of PLE against acute UV damage.

In 2019, Shalka et al. [62] evaluated the efficacy of a sunscreen SPF 90 containing 0.5% PLE in reducing sun damage. The authors recruited 10 volunteers and, on the back of each one, demarked four areas. The first was used as negative control; the second was irradiated and not pre-treated with SPF; the third was irradiated and pre-treated with SPF not containing PLE; and the last was irradiated and pre-treated with SPF supplemented by PLE. Both sunscreens had the same formulation, except for the presence of PLE. Erythema and pigmentation were evaluated colorimetrically, and, in order to perform histopathological evaluation, skin samples were collected. The areas treated with SPF 90 sunscreen containing PLE showed lower intensity of erythema and pigmentation, lower sunburn cell generation, lower MMP-1 and p53 levels, and higher CD1-a cell positivity (lower Langerhans cell depletion) compared to the areas treated with non-supplemented SPF. The authors conclude that the association of PLE with physical and chemical filters is empowered photoprotection.

The effectiveness of topical SPF in conjunction with PLE has also been investigated by other authors.

In 2021, Aguilera et al. [63] studied the effect of PLE in topical sunscreen formulations. The authors added 1% (*w*/*w*) PLE in four different galenic formulations containing different combinations of UVB and UVA organic and mineral filters. According to ISO 24443:2012, in vitro UVA-protection factor (UVA-PF), contact hypersensitivity factor (CHS), and human immunoprotection factor (HIF) were estimated. The formulation that contained PLE significantly reduced the amount of UV radiation reaching the skin. SPF and UVA-PF were significantly increased when combining UVB and UVA filters with PLE. PLE also increased UV immune protection through the elevation of CHS and HIF.

#### 6.1.3. Non-Melanoma Skin Cancer

Non-melanoma skin cancer (NMSC) includes basal cell cancer, squamous cell carcinoma, and actinic keratosis (AKs). The latter can transform into invasive squamous cell carcinoma between 0.025% and 16% of cases [64]. Nowadays, many treatments are available for AKs, including cryotherapy, topical chemotherapy, and photodynamic therapy (PDT). PDT is one of the most effective treatments, with a clearance rate that ranges from 60% to 90%. However, in approximately 20%, AKs could relapse due to the induction of local immunosuppression and DNA mutagenesis [65,66].

Assuming that PLE reduces UV-induced immunosuppression and mutagenesis, Auriemma et al. [67] in 2015 conducted a clinical study to assess the role of PLE in improving PDT efficacy and in reducing recurrences. This is the only study conducted on humans.

The authors recruited thirty-four bald patients presenting at least two AKs on the scalp and scheduled two PDT sessions one week apart. Then, they divided their population into two groups. The first one started oral PLE supplementation one week after the last PDT session at a dose of 960 mg per day for 1 month and then 480 mg per day for 5 months. The second group was followed for the same period without supplementation. All of the patients applied SPF50 sunscreen on the scalp every two hours during sun exposure. Follow-up was assessed at T1 (2 months) and T2 (6 months) with the scalp physical examination.

Both groups showed a reduction in AKs at T1 and T2 (*p* < 0.001). Nevertheless, at T2, the PDT treatment with the PLE supplementation displayed a higher clearance rate compared with PDT alone (*p* = 0.040). So, the authors concluded that PLE is a useful tool that improves PDT clearance and decreases AK recurrence. To fully understand the potential interactions between PLE and PDT, more research is needed.

In another study on mice, it was showed that PLE supplementation delayed tumor development (*p* = 0.023) [34]. The authors divided hairless mice into four groups (30 mice each): (A) non-irradiated and non-PLE-treated; (B) irradiated and non-PLE-treated; (C) non-irradiated and PLE-treated; and (D) irradiated and PLE-treated. Groups C and D received 300 mg/kg of oral PLE. Then, they exposed the mice to UVA+UVB lamps with an increasing dose from 20 to 640 mJ/cm^2^ until they presented at least a single tumor with a 4 mm diameter (at the most for 42 weeks). Therefore, they measured different parameters related to UV radiation (Langerhans cell activity, p53 expression, and Ki-67 expression).

In group D, 57% of the mice developed squamous cell carcinoma (SCC) compared to 87.5% in the untreated group and 28% developed actinic keratoses, compared to 85% in those not exposed. Additionally, it demonstrated an increased cellular relative expression of p53, suggesting that PLE can improve the mechanisms promoting DNA repair and apoptosis. Finally, in the PLE-treated mice, erythrocytic glutathione disulfide decreased, and the total plasma antioxidant capacity increased.

PLE is also useful in reducing inflammation. Long-term UV exposure could upregulate COX-2, which in turn increases prostaglandin E production, with the consequent development of epidermal hyperplasia and the loss of differentiation. The latter are involved in photodamage and carcinogenesis [5,36,59].

#### 6.1.4. Melanoma

Melanoma (MM) has undergone a great incidence increase during recent decades. There are three main risk factors for melanoma: ultraviolet radiation, genetic factors, and the presence of melanocytic nevi [31,68]. The use of topical sunscreens is the most-used method of protection against UV damage. Recently, a few oral antioxidant supplements that behave similarly to systemic photoprotective agents have appeared on the market. There are several advantages to using a systemic agent: it provides uniform, long-term, and total-body-surface protection [69].

Several mutations are implicated in MM genesis. Mutations in cyclin-dependent kinase inhibitor 2A (CDKN2A) are the most important, causing 25 to 50% of familial MM. Other genes implicated are cyclin-dependent kinase 4 (CDK4) and melanocortin 1 receptor (MC1R). The latter produces variable quantities of red/yellow pheomelanin pigment, inducing oxidative cell damage. This pigment gene, along with others (OCA, TYR), produces the red or blonde hair phenotypic (Fitzpatrick Skin Type I). These genetic alterations explain the predisposition for sun-induced freckling and low tanning response with a consequent low–moderate increased MM susceptibility risk [68,70].

In 2013, Aguilera et al. [68] conducted a clinical study to investigate PLE’s ability to decrease UV-induced erythema by analyzing the interaction between PLE and MC1R polymorphisms/CDKN2A mutations on MED. The authors recruited 61 subjects aged between 15 and 76 years old. Twenty-five had familial and/or multiple MM, 20 sporadic MM, and 16 dysplastic nevi syndrome without MM history. Each subject was assessed by phenotype (Fitzpatrick, eye color, hair color, and nevus number/description), genotype (CDKN2A and MC1R polymorphisms), and basal MED. All of the patients were treated with 720 mg of PLE in three doses plus 360 mg in a single dose 24 h and 3 h before a second MED valuation. The researcher found that oral PLE significantly increased MED in all patients (*p* < 0.05), including subjects with familiar MM (*p* < 0.05), with a higher response in women (*p* < 0.05). Furthermore, PLE showed powerful effects (assessed on the MED) in patients with familiar MM, above all in the subjects with CDKN2A mutation and/or MC1R polymorphisms. This observation did not reach a statistical significance (*p* = 0.06). Finally, PLE demonstrated a better response in patients with lower basal MEDs and dark eyes (*p* < 0.05) [68].

In 2009, a group of researchers [46] studied fibroblasts and melanoma cells and the interactions between PLE supplementation and changes in the expression of MMP and TGF-β. In cancer and photoaging, MMPs are over-expressed, while TIMPs and collagen synthesis are inhibited. Furthermore, TGF-β is inhibited in photoaging, but it is stimulated in carcinogenesis. The authors found that PLE directly inhibits MMPs and increases TIMP activities and the expression in fibroblasts and melanoma cells. Moreover, PLE stimulates collagen synthesis depending on UV exposure. Non-irradiated fibroblasts produce type I, III, and V collagen, while the irradiated ones, only I and V. Furthermore, PLE seems to stimulate TGF-β expression in non-irradiated samples, inhibiting its expression in melanoma cells.

In 2021, Serini et al. [71] conducted an in vitro study in which they assessed the antioxidant, antineoplastic, and antiaging activity of a dietary supplement containing PLE combined with sulforaphane (SFN), a sulfur-isothiocyanate present in broccoli. The PLE/SFN combination inhibited melanoma cell migration in vitro as well as MMP production, inflammasome activation, and IL-1β secretion more efficiently than individually. Furthermore, PLE/SFN was also effective in normal keratinocytes despite the presence of a pro-inflammatory environment, such as TNF-α.

### 6.2. Photoaging

Photoaging, similar to photocarcinogenesis, is strictly correlated with UVA, in particular with UVA-induced common deletion in mitochondrial DNA [40].

IN 2002, Philips et al. [32] designed a study in which fibroblasts and keratinocytes were irradiated by a single exposure to UVA or UVB. The samples were incubated with PLE at different concentrations (0.01, 0.1, and 1%). All of the cells were examined for membrane damage, lipid peroxidation, the expression of elastin (protein levels), and MMP-1 (protein levels or MMP-1 promoter activity). The authors found that lower concentrations of PLE (lower than 0.1%) may prevent photoaging, improving membrane integrity and inhibiting MMP-1. Moreover, higher concentrations (greater than 0.1%) may reverse elastic fiber loss.

In 2010, Villa et al. [40] assessed the PLE effects on common deletions in fibroblast and keratinocytes due to chronic UVA radiation in 10 patients. All of the healthy volunteers underwent a first 3 mm punch biopsy to obtain baseline histology. Half of the group was treated with PLE (240 mg 8 and 2 h before exposure), and then, all of the subjects were exposed to artificial UVA light at a dose of 10, 15, 20, 25, 30, or 35 J/cm^2^. After 24 h, each patient underwent two punch biopsies from the irradiated areas and one punch biopsy from an adjacent nonirradiated site. The authors described fewer common deletions in patients treated with PLE, suggesting that it might prevent UVA-induced photodamage, preventing UVA-dependent mitochondrial DNA damage. These results did not reach statistical significance, probably due to sample size [29].

As well as UVR, IR and visible light (VIS) can also cause erythema, hyperpigmentation, genotoxicity, or MMPs [72].

Zamarron et al. [72] evaluated the protective role of PLE against infrared A (IRA) and VIS radiation in human dermal fibroblasts. The researchers examined PLE effects on morphology, viability, cell cycle, and extracellular matrix expression. It appears that PLE protects cells from the damage caused by VIS and IRA. Moreover, it inhibits the increase in MMP-1 and cathepsin K induced by VIS and IRA as well as the changes in fibrillin 1, fibrillin 2, and elastin levels.

Additionally, blue light induces oxidative stress, long-lasting pigmentation, and photoaging [73].

In 2021, Portillo et al. [73] studied how PLE decreases the hyperpigmentation induced by blue light from digital devices. The authors exposed human dermal fibroblasts (HDF) and murine melanocytes (B16-F10) to artificial blue light (a 400–500 nm LED lamp). Then, they examined cell viability, mitochondrial morphology, and the expression of MAPK- p38 (pathways involved in melanogenesis). Furthermore, they examined the activation of Opsin-3, a membrane protein sensitive to blue light that activates melanogenesis enzymes in melanocytes. According to the researcher, PLE (0.1, 0.3, and 0.5 mg/mL for 24 h before irradiation) prevents cell death, the alteration of mitochondrial morphology, and the phosphorylation of p38 in HDF exposed to blue light. Additionally, PLE significantly reduces melanocyte Opsin-3 activation, preventing melanin photooxidation and photodegradation. In conclusion, PLE provides protection from the harmful effects of the blue light of digital devices, as well as preventing early photoaging.

In 2020, Delgado-Wicke et al. [74] found that air pollution acts in a synergic way with UV radiation in order to cause photoaging. Starting from the assumption that nuclear factor erythroid 2–related factor 2 (NRF2) controls an important anti-oxidant pathway [75], the authors found that PLE elicits NRF2-dependent antistress responses. In particular, PLE, increasing the NRF2 pathway, is effective against UV-induced oxidative stress, but also against the xenotoxic oxidative stress produced by exposure to fine particulate pollutants (e.g., PM2.5).

## 7. Other Uses

PLE proved its efficacy in other important diseases, such as primary or autoimmune photodermatosis [69,76,77], pigmentation disorders such as vitiligo [78,79,80,81,82] and melasma [5,83,84,85], atopic dermatitis [52], subacute cutaneous lupus erythematosus (SCLE) [86], and porphyria cutanea tarda [87].

## 8. Conclusions

UVR is involved as a causative or worsening agent in the pathogenesis and in the clinical history of several cutaneous diseases. Growing evidence suggests the efficacy and safety of PLE in the treatment and prevention of several cutaneous diseases, such as NMSC, melanoma, disorders of pigmentation, photosensitivity, and atopic dermatitis.

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
