# Peer review of "Uses of Polypodium leucotomos Extract in Oncodermatology"

_jcm, 2023, doi:10.3390/jcm12020673_

Round 1
Reviewer 1 Report
In line 184, 382 an H is missing in Hiperpigmentation
Reference 51 is missing the name of the authors
Author Response
Dear Reviewer,
thank you for your positive comments about our paper.
Q: In line 184, 382 an H is missing in Hiperpigmentation
A: Thank you, the orthographic mistake has been corrected
Q: Reference 51 is missing the names of the authors
A: Thank you, we added the names of the authors
Kind regards
Reviewer 2 Report
The authors of "Uses of Polypodium leucotomos extract in oncodermatology: a 2 review have provided a well organized review with comprehensive description on how polypodium leucotomos is used in oncodermatology as well as non neoplastic conditions.
The review is very informative and delivers the information systematically. It still requires some editing when it comes to complexity of some sentences and run on sentences.
Author Response
Dear Reviewer,
thank you for the positive comment about our paper. Now, the manuscript has been completely reviewed by a native English-speaking author.
Kind regards
Reviewer 3 Report
Comments:
Abstract
It needs to be edited
Introduction
Page 1 of 14:
1. Line 25. “Ozone screens out significant quantities of UVB and UVC. Therefore, solar UV radiation…”
Before this phrase, although it is repetitive, a phrase that indicates that UV light is divided into three components (A, B and C) depending on the wavelength would be recommended.
2. Line 35 “UVA radiation induces formation of 8-deoxyguanosine (8-OH-dG)[3], a product of DNA oxidation…” edition
Page 2 of 14:
3. Line 45 “…Also, UVR ARE able to act on Langerhans cells and mastocytes…” grammatical error
Origin and chemical composition of Polypodium leucotomos
4. Line 82 Título: “2. Origin and chemical composition of Polypodium leucotomos”
This section refers to PLE, that is, to the origin and composition of the extract used as a photoprotector, not to PL, which is the abbreviation that has been given to refer to the fern itself.
5. Line 29 Locally it is known as “calaguala” and extracts are called "anapsos"[29].
This statement is wrong and needs to be corrected. Only the product used in Atopic Dermatitis (pur ref 54) is made with anapsos, the other extracts are not known as anapsos and therefore the composition is different. A work published in the J Med Plants Res (your ref 26) tested several of them and it was verified by HPLC that they had different profiles and their biological activities were different or not completely similar, while some had photoprotective properties and others did not. By the way, this reference is wrongly written.
Pharmacodynamic
Page 3 of 14:
6. Line 17 “This antioxidant activity of PLE was demonstrated by the reduction of the levels 8-OH-dG, a marker of cellular oxidative damage” Has the 8-OH-dG test been the only analysis or evaluation indicating the antioxidant capacity of PLE?
7. Line 130 “COX-2 is responsible for prostaglandin synthesis ….. also involved in carcinogenesis., too”
Why mention COX2 here, what about PL on this enzyme?
Page 4 of 14:
Formulation, dosage and safety
8. Line 152 “In the current formulation, it is marketed as a dietary supplement aimed at "protecting against sun-related effects and aging", with a recommendation of one capsule of 240 mg in the morning, and one capsule of 240 mg a day when extensive sun exposure is anticipated.” This dosage didn't work for me. I have searched the original article from which the information is taken Authors should correct it.
9. Line 147 “Furthermore, a study in vitro demonstrated that the best antioxidant and photoprotective activity is obtained with an aqueous PLE extracted at basic pH”
Is this extract another type of formulation? Or are you referring to an extract for topical use? If it is only tested in vitro, perhaps this information should be moved to another section, not this one where clinically approved formulations without side effects are specified.
Clinical use in photodamage prevention
10. Line 167 “Skin cancer is primarily caused by UV radiation exposure and its pathogenesis is strictly correlated to UVA”
Only to UVA, please edit
Page 5 of 14:
11. “An extract of hydrophilic PLE was used to pre-treat these cells and then UV rays were used to irradiate them”
Which UV component was used for irradiation, A or B?
12. Line 202 “…the first one was feeded with PLE 300 mg/kg for 10 days, the second one didn’t receive drugs. Then, they irradiated mice with UV one time”
What type of UV was used: A or B? What dose was applied or how long were they irradiated?
13. Line 203 “The Authors assessed the reduction of the COX-2 expression, p53 expression and the reduction of cyclobutane pyrimidine photoproducts.
Authors should edit this sentence p53 is reduce or increased
14. “Therefore, PLE reduced UV-induced COX-2 levels in mice, by promoting p53 expression, decreasing prostaglandin-mediated inflammation linked to photoaging.”
Here the same information is repeated at the beginning as in the previous paragraph (COX-2 reduction, p53 promotion), but new information is added regarding prostaglandins and inflammation that I don't know if it also comes from the Zattra study or if It is from another study, it should be clarified.
15. Line 213 Kholi et al.: “Subjects were irradiated with visible light, ultraviolet A1 (UVA1), and ultraviolet B (UVB) using a 308-nm excimer laser on day 1. A clinical assessment was conducted immediately after irradiation and 24 hours later. Afterward, on days 3 and 4, patients were given 240 mg of PLE 2 hours and 1 hour before irradiation, followed by a repeated evaluation. On day 2 and 4, skin biopsies were performed.” In general, the sequence of this study is worded in a confusing way: what nm does UVA1 correspond to? Does the laser at 308 nm refer only to UVB radiation? Then the light sources of the other radiations would have to be explained. The methodology of PLE irradiations and administration is not well understood.
16. Line 224 Portillo et al.:
- At the beginning of the paragraph it is indicated that the study was carried out in mice, Are you sure????.
17. Line 233 “Torricelli et al.[62] conducted an in vitro study to evaluate the effects of topical PLE on UVB- induced cell damage. Topical application of 2mg/cm2 of PL, immediately before UVB exposure, reduced sunburn cells (80%), p53 (80%), p21 (84%), and ki-67 (48%) positive cells, as well as cyclopyrimidine dimers (CPD), confirming the photoprotective effect of PLE against acute UV damage.” En amarillo: What cell type was used? How was the reduction of sunburn cells evaluated in this in vitro study? It seems that the authors confuse the model since they indicate that the PL was applied at doses in mg/cm2
Page 6 of 14:
18. Line 290 “In another study on mice, PLE supplementation delayed tumor development (P = 285 .023)[34]. The Authors divided hairless mice into two groups. Group 1 received oral PLE (300 mg/kg), whereas group 2 other did not. They then exposed mice to UVA/UVB lamps for 42 weeks and they measured different parameters related to UV radiation (Langerhans 288 cell activity, p53 expression, and Ki-67 expression).
:- Was UVA, UVB or both radiations used? If they were combined, how were the doses?
- Indicates that mice were exposed to UV light for 42 weeks. This is not well explained. I understand it was a 42 week chronic UV exposure experiment, but how often did the irradiation occur at what dose/power?
Page 8 of 14:
19. Line 361 “Villa et al. in 2010[40] assessed ……….. patient had 2 punch biopsies taken 24 hours later from the two irradiated areas and a fourth biopsy taken from an adjacent nonirradiated site. These results didn’t reach statistical significance, probably due to simple size, but the Authors described fewer common deletion in patients treated with PL, suggesting that PLE might prevent UVA-induced photodamage, maybe preventing UVA-dependent mitochondrial DNA damage.”
First: the patients were “healthy volunteers”? Or did they have some developed pathology?; Second: What dose of UVA was applied? Was it chronic, as the first sentence of the paragraph cites?; Third: the ending is unclearly worded. There is talk of 2 biopsies and after a fourth... and the third? Then it is said that the results were not statistically significant, but the results are not described prior to saying that.
Page 9 of 14:
Other uses
20. Line 403 “PLE prove its efficacy in others important disease such as primary or autoimmune photodermatosis [70,78,79], pigmentation disorders like vitiligo[80–84] and melasma[5,85–87], atopic dermatitis[54], subacute cutaneous lupus erythematosus (SCLE)[88], porphyria cutanea tarda[89].”
Typos/grammatical errors
Conclusions
21. Line 408 “UVR are involved as causing or worsening agents in the pathogenesis and in the clinical history of several cutaneous diseases. Growing evidence suggests the efficacy and the safety of PLE in the treatment and in the prevention of several cutaneous diseases, such as NMSC, melanoma, disorders of pigmentation, photosensitivity and atopic dermatitis.”
Typos/grammatical errors
General considerations
- To homogenize the nomenclatures. I would put the developed name of all the markers/proteins/genes named.
- To homogenize the form of include bibliographical quotes in the text. example: “Middelkamp-Hup[58] in 2004…”, “In 2007, JanczykIn et al. performed an in vitro…..[39].” Either the reference is included next to the author in all of them (as in the first case) or it is included at the end of the explanatory paragraph of the study (as in the second case).
- In general, in all the sections, I would order the studies by putting the in vitro studies first, then the in vivo studies in animal models and, lastly, the clinical trials in humans (if any).
- Even more important, To improve the section Origin and chemical composition of Polypodium leucotomos indicating that the PL extract used in >90% of the articles corresponds to an aqueous extract of PL leaves, since the combination of ingredients depends on it, I would even recommend the patent reference of the product
Author Response
Dear Reviewer,
thank you for your detailed review and valuable advice. These are our replies.
Q: Line 25. “Ozone screens out significant quantities of UVB and UVC. Therefore, solar UV radiation…”
Before this phrase, although it is repetitive, a phrase that indicates that UV light is divided into three components (A, B and C) depending on the wavelength would be recommended.
A: Thank you, the sentence concerning the subdivision of UVR has been added.
Q: Line 35 “UVA radiation induces formation of 8-deoxyguanosine (8-OH-dG)[3], a product of DNA oxidation…” edition
A: Thank you, the sentence has been edited as suggested.
Q: Line 45 “…Also, UVR ARE able to act on Langerhans cells and mastocytes…” grammatical error
A: Thank you, the grammatical error has been corrected.
Q: Line 82 Título: “2. Origin and chemical composition of Polypodium leucotomos”
This section refers to PLE, that is, to the origin and composition of the extract used as a photoprotector, not to PL, which is the abbreviation that has been given to refer to the fern itself.
A: Thank you, the title has been edited.
Q: Line 29 Locally it is known as “calaguala” and extracts are called anapsos[29].
This statement is wrong and needs to be corrected. Only the product used in Atopic Dermatitis (pur ref 54) is made with anapsos, the other extracts are not known as anapsos and therefore the composition is different.
A work published in the J Med Plants Res (your ref 26) tested several of them and it was verified by HPLC that they had different profiles and their biological activities were different or not completely similar, while some had photoprotective properties and others did not. By the way, this reference is wrongly written.
A: Thank you, this statement has been deleted to avoid misunderstandings. The reference has been checked and it results correct.
Q: Line 17 “This antioxidant activity of PLE was demonstrated by the reduction of the levels 8-OH-dG, a marker of cellular oxidative damage” Has the 8-OH-dG test been the only analysis or evaluation indicating the antioxidant capacity of PLE?
A: Thank you, in this context the 8-OH-dG test was the main analysis conducted to show the antioxidant capacity of PLE.
Q: Line 130 “COX-2 is responsible for prostaglandin synthesis ….. also involved in carcinogenesis., too” Why mention COX2 here, what about PL on this enzyme?
A: Thank you, a statement has been added to clarify the role of PLE on the enzyme.
Q: Line 152 “In the current formulation, it is marketed as a dietary supplement aimed at "protecting against sun-related effects and aging", with a recommendation of one capsule of 240 mg in the morning, and one capsule of 240 mg a day when extensive sun exposure is anticipated.” This
dosage didn't work for me. I have searched the original article from which the information is taken Authors should correct it.
A: Thank you, the statement has been corrected.
Q: Line 147 “Furthermore, a study in vitro demonstrated that the best antioxidant and photoprotective activity is obtained with an aqueous PLE extracted at basic pH”
Is this extract another type of formulation? Or are you referring to an extract for topical use? If it is only tested in vitro, perhaps this information should be moved to another section, not this one where clinically approved formulations without side effects are specified.
A: Thank you, the sentence about the study in vitro has been moved to a previous sections.
Q: Line 167 “Skin cancer is primarily caused by UV radiation exposure and its pathogenesis is strictly correlated to UVA” Only to UVA, please edit
A: Thank you, the mistake has been corrected
Q: “An extract of hydrophilic PLE was used to pre-treat these cells and then UV rays were used to irradiate them”. Which UV component was used for irradiation, A or B?
A: Thank you, the sentence concerning the UV component has been added.
Q: Line 202 “…the first one was feeded with PLE 300 mg/kg for 10 days, the second one didn’t receive drugs. Then, they irradiated mice with UV one time”
What type of UV was used: A or B? What dose was applied or how long were they irradiated?
A: Thank you, the sentence concerning the UV component and the dose has been added.
Q: Line 203 “The Authors assessed the reduction of the COX-2 expression, p53 expression and the reduction of cyclobutane pyrimidine photoproducts. Authors should edit this sentence p53 is reduce or increased
A: Thank you, the mistake has been corrected
Q: “Therefore, PLE reduced UV-induced COX-2 levels in mice, by promoting p53 expression, decreasing prostaglandin-mediated inflammation linked to photoaging.” Here the same information is repeated at the beginning as in the previous paragraph (COX-2 reduction, p53 promotion), but new
information is added regarding prostaglandins and inflammation that I don't know if it also comes from the Zattra study or if It is from another study, it should be clarified.
A: Thank you, the information regarding prostaglandins and inflammations comes from Zattra study.
Q: Line 213 Kholi et al.: “Subjects were irradiated with visible light, ultraviolet A1 (UVA1), and ultraviolet B (UVB) using a 308-nm excimer laser on day 1. A clinical assessment was conducted immediately after irradiation and 24 hours later. Afterward, on days 3 and 4, patients were given 240 mg of PLE 2 hours and 1 hour before irradiation, followed by a repeated evaluation. On day 2 and 4,
skin biopsies were performed.” In general, the sequence of this study is worded in a confusing way: what nm does UVA1 correspond to? Does the laser at 308 nm refer only to UVB radiation? Then the light sources of the other radiations would have to be explained. The methodology of PLE irradiations and
administration is not well understood.
A: Thank you, the sentence has been edited as suggested.
Q: Line 224 Portillo et al.: At the beginning of the paragraph it is indicated that the study was carried out in mice, Are you sure????.
A: Thank you, the statement has been corrected. The study was performed in vitro.
Q: Line 233 “Torricelli et al.[62] conducted an in vitro study to evaluate the effects of topical PLE on UVB- induced cell damage. Topical application of 2mg/cm2 of PL, immediately before UVB exposure, reduced sunburn cells (80%), p53 (80%), p21 (84%), and ki-67 (48%) positive cells, as well as cyclopyrimidine dimers (CPD), confirming the photoprotective effect of PLE against acute UV
damage.” En amarillo: What cell type was used? How was the reduction of sunburn cells evaluated in this in vitro study? It seems that the authors confuse the model since they indicate that the PL was applied at doses in mg/cm2.
A: Thank you, the statement has been corrected.
Q: Line 290 “In another study on mice, PLE supplementation delayed tumor development (P = 285.023)[34]. The Authors divided hairless mice into two groups. Group 1 received oral PLE (300 mg/kg), whereas group 2 other did not. They then exposed mice to UVA/UVB lamps for 42 weeks and they
measured different parameters related to UV radiation (Langerhans 288 cell activity, p53 expression, and Ki-67 expression). - Was UVA, UVB or both radiations used? If they were combined, how were the doses?- Indicates that mice were exposed to UV light for 42 weeks. This is not well explained. I understand it was a 42 week chronic UV exposure experiment, but how often did the irradiation occur at what dose/power?
A: Thank you, the statement has been corrected.
Q: Line 361 “Villa et al. in 2010[40] assessed ……….. patient had 2 punch biopsies taken 24 hours later from the two irradiated areas and a fourth biopsy taken from an adjacent nonirradiated site. These results didn’t reach statistical significance, probably due to simple size, but the Authors described fewer common deletion in patients treated with PL, suggesting that PLE might prevent
UVA-induced photodamage, maybe preventing UVA-dependent mitochondrial DNA damage.” First: the patients were “healthy volunteers”? Or did they have some developed pathology?; Second: What dose of UVA was applied? Was it chronic, as the first sentence of the paragraph cites?; Third: the ending is unclearly worded. There is talk of 2 biopsies and after a fourth... and the third? Then it is said that the results were not statistically significant, but the results are not described prior to saying that.
A: Thank you, the statement has been corrected.
Q: Line 403 “PLE prove its efficacy in others important disease such as primary or autoimmune photodermatosis [70,78,79], pigmentation disorders like vitiligo[80–84] and melasma[5,85–87], atopic dermatitis[54], subacute cutaneous lupus erythematosus (SCLE)[88], porphyria cutanea tarda[89].”
Typos/grammatical errors
A: Thank you, the grammatical error has been corrected.
Q: 21. Line 408 “UVR are involved as causing or worsening agents in the pathogenesis and in the clinical history of several cutaneous diseases. Growing evidence suggests the efficacy and the safety of PLE in the treatment and in the prevention of several cutaneous diseases, such as NMSC, melanoma, disorders of pigmentation, photosensitivity and atopic dermatitis.” Typos/grammatical
errors.
A: Thank you, the grammatical error has been corrected.
Finally, the patent reference of Fernblock has been added as you suggested.
Kind regards
Round 2
Reviewer 3 Report
I just want to indicate that after reviewing the manuscript and answers to my comments I was very glad with them and with the revised manuscript. I don´t have any additional comments